# Transmembrane and Tetratricopeptide Repeat Containing 4 Is a Novel Diagnostic Marker for Prostate Cancer with High Specificity and Sensitivity

**DOI:** 10.3390/cells10051029

**Published:** 2021-04-27

**Authors:** Rania Makboul, Islam F. Abdelkawi, Dalia M. Badary, Mahmoud R. A. Hussein, Johng S. Rhim, Eman A. Toraih, Mourad Zerfaoui, Zakaria Y. Abd Elmageed

**Affiliations:** 1Department of Pathology and Urology, Faculty of Medicine, Assiut University, Assiut 71111, Egypt; raniamakboul@aun.edu.eg (R.M.); daliabadary@aun.edu.eg (D.M.B.); frcpath17@gmail.com (M.R.A.H.); 2Department of Urology, Faculty of Medicine, Assiut University, Assiut 71111, Egypt; islamfaa@yahoo.com; 3Department of Surgery, Uniformed Services University of the Health Sciences, Bethesda, MD 20814, USA; jrhim@verizon.net; 4Department of Surgery, Tulane University School of Medicine, 1430 Tulane Avenue, New Orleans, LA 70112, USA; etoraih@tulane.edu (E.A.T.); mzerfaoui@tulane.edu (M.Z.); 5Department of Pharmacology, Edward Via College of Osteopathic Medicine, University of Louisiana at Monroe, Monroe, LA 71203, USA

**Keywords:** prostatic cancer, TMTC4, immunohistochemical staining, biomarker, sensitivity and specificity

## Abstract

The histopathologic diagnosis of prostate cancer (PCa) from biopsies is a current challenge if double or triple staining is needed. Therefore, there is an urgent need for development of a new reliable biomarker to diagnose PCa patients. We aimed to explore and compare the expression of TMTC4 in PCa cells and tissue specimens and evaluate its sensitivity and specificity. The expression of TMTC4 in PCa and normal prostate epithelial cells was determined by real-time PCR and Western blot analyses. Immunohistochemical (IHC) staining of TMTC4 was performed on tissues collected from PCa and benign prostatic hyperplasia (BPH). Our results show a high expression of TMTC4 on mRNA and protein levels in PCa versus BPH1 and normal cells (*p* < 0.05). IHC results show strong cytoplasmic expressions in PCa cases (*p* < 0.001) as compared to BPH cases. The overall accuracy as measured by the AUC was 1.0 (*p* < 0.001). The sensitivity and specificity of the protein were 100% and 96.6%, respectively. Taken together, we report a high TMTC4 expression in PCa cells and tissues and its ability to differentiate between PCa and BPH with high sensitivity and specificity. This finding can be carried over to clinical practice after its confirmation by further studies.

## 1. Introduction

Prostate cancer (PCa) is the second most commonly diagnosed male malignancy in the United States with more than 248,530 cases that will be diagnosed and 34,130 estimated deaths in 2021 [1]. Although PCa can be diagnosed early and adequately treated, some tumors do still recur and progress aggressively, including androgen dependent and independent pathways [2]. Although prostate-specific antigen (PSA) is a routine diagnostic marker for PCa, its sensitivity is compromised by false positive results, which might lead to overtreatment of indolent disease [3] and mislead treatment decisions at relapse [4]. Faced with these diagnostic and clinical situations, the finding of novel and reliable clinical biomarkers remains important [5,6].

Although several histological diagnostic criteria are described for PCa diagnosis, these have certain limitations. Moreover, most of these criteria may also be present in benign mimickers. This explains the challenge of diagnosis of PCa, especially in needle biopsy. In addition, some unusual variants of PCa, such as foamy carcinoma, have bland nuclear features. However, inflamed benign glands, atrophy, and atypical basal cell hyperplasia can occasionally display features seen in PCa. One of the most important criteria in diagnosing PCa is the absence of basal cells around the malignant glands [7]. Morphological distinction of basal cells from stromal fibroblasts is often difficult using H&E staining. Furthermore, basal cells may be inconspicuous in some benign glands. Thus, immunohistochemical confirmation of PCa is necessary in certain cases and is traditionally based on the immunoreactivity of basal cell markers (p63 and CK5/6) and cytoplasmic expression of α-methylacyl-CoA racemase (AMACR) [7]. 

A family of four homologous genes that encode transmembrane and tetratricopeptide repeat (TPR) containing protein identified as TMTC1-4 was discovered by large human sequencing studies [8,9]. The fate and function of many cellular processes in normal and abnormal cells depend on the number of tetratricopeptide repeats clusters and their ability to bind to specific types of ligand proteins. At least three consecutive TPR domains are required to form a functional unit of the protein [10]. Although the exact functions and molecular pathways related to these molecules are still to be uncovered, few studies suggested that they have roles in the endoplasmic reticulum (ER) function, unfolded protein response (UPR), and pathways related to cadherins (CDHs) and protocadherins (PCDHs) [11,12,13,14]. Recently, the role of these pathways in carcinogenesis is being explored and targeted [14,15,16]. Therefore, the identification of new regulatory molecules may have possible diagnostic and therapeutic implications. To the best of our knowledge, the assessment of TMTC protein expression in PCa has not yet been performed.

This study aimed to explore whether TMTC4 expression has a diagnostic role in PCa through the evaluation of its expression patterns at various grades of the disease and comparison with the expression of the protein in tissues collected from patients with benign hyperplasia.

## 2. Results

### 2.1. Upregulation of TMTC4 in Prostate Cancer Cells

In the current study, cell lines and human tissue samples collected from 72 patients were utilized to investigate the expression of TMTC4, as depicted in Figure 1A. We initiated our study by evaluating the expression of *tmtc3* and *tmtc4* on the level of mRNA in a panel of cells including PCa, BPH1 and normal prostatic epithelial PrEC and immortal normal RWPE1 cells. The results reveal that the two transcripts were upregulated in BPH1 cells compared to normal cells (*p* < 0.05). In comparison to BPH1 cells, *tmtc3* transcript was upregulated (*p* < 0.05) in most of the PCa cells except LNCaP cells where *tmtc4* was markedly upregulated in the studied PCa cells (*p* < 0.001), as shown in Figure 1B. Thus, our next experiments focused on only TMTC4 protein expression in cells and human tissues. The differential expression of the *tmtc4* transcript was confirmed on protein level. Western blot analysis showed that TMTC4 was overexpressed in BPH1(*p* < 0.05) compared to RWPE1 cells and in RC77 T/E (*p* < 0.05) compared to RC77 N/E cells (Figure 1C). Compared to benign hyperplastic BPH1 cells, TMTC4 was highly expressed (*p* < 0.01) in Du145 cells. The complete gel of Western blot analysis is provided in Appendix A.

### 2.2. Immunohistochemical Staining of TMTC4 in BPH and PCa Tissue Specimens

Of the 29 PBH cases, five cases were associated with chronic prostatitis, a single case was associated with acute prostatitis, and another one was associated with basal cell hyperplasia. According to ISUP, the PCa cases were Grades 2 (*n* = 10), 3 (*n* = 4), 4 (*n* = 10), and 5 (*n* = 19). Perineural invasion was detected in 37.2% (16/43) of cases, as depicted in Table 1. As shown in Figure 2 and Figure 3, the expression of TMTC4 was unequivocally cytoplasmic. A strong protein signal was observed in tissues procured from PCa patients compared to a weak signal detected in BPH tissues. The mean expression of TMTC4 was 278.12 ± 41.54 in PCa versus 16.69 ± 34.4 in BPH tissues (*p* < 0.001), as shown in Table 2. Interestingly, there was no significant difference in TMTC4 expression among PCa cases with regards to tumor grades using Gleason or ISUP grading system (Figure 4B), age below or above 65 years (Figure 4C), or the presence or absence of perineural invasion (Figure 4D). 

### 2.3. TMTC4 Expression Discriminates PCa from BPH Patients with High Accuracy

ROC curve was depicted for TMTC4 protein expression in PCa and BPH patients, and the overall accuracy measured by the AUC was 1.0 when the cut-off point of H-score was ≥100 (*p* < 0.001), as illustrated in Table 3 and Figure 4E. The sensitivity and specificity of the TMTC4 expression for discriminating PCa from BPH tissue specimens were 100% and 96.6%, respectively. The positive predictive value of the protein was 97.7% and negative predictive value was 100%.

### 2.4. Upregulation of tmtc4 Transcript in an Independent Cohort of PCa

We started to look at the expression of TMTC4 in an independent cohort and a large number of PCa samples. TCGA data were retrieved and analyzed for 497 localized prostate tumors in addition to 52 normal prostate tissues, as previously described [17]. Confirming our results from PCa cell lines and human specimens, *tmtc4* transcript was significantly upregulated in tissues collected from PCa patients compared to normal healthy subjects (Figure 5A). This expression pattern was also observed in PCa tissues for Gleason scores (GS) of 6–9 (*p* < 0.001) with regard to normal tissues, as shown in Figure 5B. Regarding the molecular signature of PCa tissues, the transcript was upregulated in those tissues which have ERG, ETV1, and ETV4 fusion and SPOP mutations (*p* < 0.01), as depicted in Figure 5C. Although GS 10 and other molecular features had no significant differences, the number of patients is comparably limited. Further studies are required to address these genetic factors using a reasonable number of PCa tissues.

## 3. Discussion

Although PSA is the standard biomarker used in PCa diagnosis and prognosis, false positive results limit its use and increase the need for new marker discovery. When patients with benign hyperplasia or prostatitis are screened for PSA, false positive results are usually expected [18]. As a result, a large fraction of patients must undergo unnecessary biopsies which cause discomfort, increase healthcare cost, and have psychological consequences [19]. To address this clinical unmet need, we evaluated TMTC4 expression in PCa cells and human tissues and compared it with normal and BPH cells using real-time PCR, Western blot, and IHC analyses. In PCa cells, *tmtc3* and *tmtc3* transcripts were upregulated regarding normal and BHP1 cells. On the protein level, TMTC4 was significantly expressed in BPH1 versus normal cells and DU145 versus BPH1 cells. This expression pattern was recapitulative in PCa RC77 T/E cells when compared to their matched normal RC77 N/E cells. Recently identified renal cell carcinoma E006AA/hT cells [20] were used as a control for TMTC4 expression. In human tissues, TMTC4 protein expression was able to differentiate between PCa and BPH with high sensitivity and specificity. When we applied ROC curve, we used H-score at a cut-off of ≥100, and the results show that the sensitivity and specificity were 100% and 96.6%, respectively, and the positive and negative predictive values were 97.7% and 100%, respectively. As a result, we advocate the validation of TMTC4 H-score of at least 100 be used as a cutoff in future studies. According to the ISUP recommendations, the need for a double/triple-antibody staining, which is recommended for diagnosis of PCa in some cases, may be countered by the fact that small foci may not survive sectioning to perform the staining [21]. Of note, if we could use TMTC4 antibody alone to differentiate between benign and malignant prostatic lesions, it would hold promise in PCa diagnosis. Although our results may have an important clinical application, there is a need for further confirmation by other multicenter studies.

It was reported that TMTC4 is a member of a family consisting of four proteins, the exact functions of which are still not fully understood, but, as the sequence analysis predicted similar overall protein products, they likely share overlapping functions. They are mainly located in the endoplasmic reticulum (ER) and have roles in protein folding and calcium regulation [11]. Since cancer cells are under significant stress, they rely on adaptive responses for survival. These suggested mechanisms include ER stress response and the unfolded protein response (UPR) [15]. These pathways have been considered in both hormone-sensitive and castration-resistant PCa and are being therapeutically targeted [22]. The fate and function of many cellular processes in normal and abnormal cells depend on the number of TPR clusters and their ability to bind to a specific type of ligand proteins. TMTC2 protein belongs to the same family, binds to the calcium uptake pump ER calcium ATPase 2b (SERCA2b) and the carbohydrate-binding chaperone calnexin, and appears to play a potential role in calcium homeostasis in the ER. It is thought to control ER Ca^2+^ dynamics and its downstream ER UPR, which, when experimentally deactivated in mice, resulted in cochlear hair cell death [12]. The amount of available Ca^2+^ regulates most physiological processes inside the cell, including cell proliferation and apoptosis. SERCA is one of these Ca^2+^ regulators, which controls the cellular Ca^2+^ efflux process from the cytosol to the ER [23]. It was reported that dysregulation of Ca^2+^ signaling is working in the favor of cancer cells [24] and overexpression of SERCA activates signaling pathways associated with cell survival [25]. Overexpression of SERCA2b reduced curcumin-induced PARP1 cleavage in ovarian cancer cells, suggesting its role in the development of ovarian cancer [25]. This may be an explanation for the TMTC4 overexpression in tumor cells. The other possible link between TMTC proteins and PCa is the posttranslational modification of cadherins, a process that is considered crucial in human cancers [26]. However, the mechanism that regulates E-cadherin functions in malignant cells through posttranslational modification and the role of the newly described *O*-mannosylation needs further investigation [14].

Our current study has compelling findings because: (1) the high protein expression of TMTC4 in human tissue specimens was confirmed in PCa as compared to BPH tissue samples with high sensitivity and specificity; (2) the protein expression of TMTC4 was strong in all grades of PCa; and (3) there is the possibility of using this protein in PCa biopsies as a unique marker. However, the limitations of our study include: (1) the number of cases under study is limited, thus extended study with more cases is recommended; (2) the expression of this protein in premalignant conditions such as prostatic intraepithelial neoplasia (low and high PIN) was not investigated; (3) its expression in other PCa mimickers should be explored to confirm its utility as a diagnostic marker for PCa; and (4) the protein expression should be tested in various histologic types of PCa, especially those which are difficult to diagnose due to the histological similarity of benign conditions.

## 4. Materials and Methods

### 4.1. Patients and Specimens

Prior to initiating human subjects research, informed consent was obtained from patients included in this study after Institutional Review Board (IRB)-approval by the Faculty of Medicine, Assiut University, Assiut. The research work involving humans was carried out in accordance with the Code of Ethics of the World Medical Association (Declaration of Helsinki). Tissue samples from 72 randomly selected consecutive patients were obtained from the archives of the Department of Pathology, Assiut University Hospital. The samples were reviewed by two expert pathologists to confirm the histopathologic diagnosis of benign prostatic hyperplasia (BPH) and prostate cancer (PCa) in 29 and 43 cases, respectively, and representative formalin-fixed paraffin-embedded blocks were selected for immunohistochemical studies. PCa cases were graded according to Gleason score and ISUP grading systems [27].

### 4.2. Cell Culture

The human PCa PC-3, Du145, and C4-2B cells as well as BPH1 cells were obtained from the American Type Culture Collection (ATCC, Manassas, VA, USA) and maintained in DMEM medium containing 10% FBS and 1% penicillin/streptomycin. RWPE-1, RC77T/E, and RC77N/E cells were grown in keratinocyte serum-free medium (K-FSM) supplemented with bovine pituitary extract and EGF following the manufacturer’s instructions (Invitrogen, Carlsbad, CA, USA) and as described [28]. Cell lines were examined every six months for mycoplasma using PCR kit (ATCC, Manassas, VA, USA). 

### 4.3. Gene Expression Analysis in PCa Cells 

Total RNA from PCa and normal cells was extracted using Trizol reagent according to the manufacturer’s protocol (Invitrogen Corp., Carlsbad, CA, USA). cDNA was prepared and quantitative real-time PCR was performed using SYBR Green master mix (Bio-Rad, Hercules, CA, USA) on a Bio-Rad CFX96 Touch^TM^ detection system. The following primer sequence was used: *tmtc3* 5′-TTTTCCTAAGCCATCCCCTG-3′ and 5′- CAAAACCACAAAAGAGGCTG-3′, *tmtc4* 5′-CCCTCATTAAGTCCATCAGCG -3′ and 5′-ATAACGAGAAATCCCAGGCC-3′, and *gapdh* 5′-GAGTCAACGGATTTGGTCGT-3′ and 5′-TTGATTTTGGAGGGATCTCG -3′. The fold change of gene expression was calculated relative to *gapdh* by comparing the Ct method as described [29].

### 4.4. Western Blot Analysis

Western blot was performed as we described [30]. Briefly, 30 µg of protein lysate were fractionated, transferred onto a nitrocellulose membrane, and blocked with 5% BSA. The membranes were incubated overnight at 4 °C with primary TMTC4 antibody (Y-19), Santa Cruz Biotechnology, Dallas, TX, USA) followed by 1 h secondary antibody incubation at room temperature and visualized by chemiluminescence (Pierce, Rockford, IL, USA) on C-Digit Blot Scanner (LI-COR Biosciences, Lincoln, NE, USA).

### 4.5. Immunohistochemical Staining

The protein expression of TMTC4 was assessed by immunohistochemical (IHC) staining using avidin biotin immunoperoxidase complex technique (UltravisionPlus Detection System anti-polyvalent HRP/DAB, ready to use; Thermo Scientific Co., Fremont, CA, USA). The steps of IHC were performed according to the manufacturing protocol. Tissue sections were deparaffinized, rehydrated, and immersed in citrate buffer and then boiled for 30 min. Sections were incubated overnight at room temperature with diluted primary TMTC4 antibody (TMTC4 (Y-19), Santa Cruz Biotechnology, Dallas, TX, USA) at a dilution of 1:50. After the application of secondary antibody, diaminobenzidine was applied and the sections were then counterstained with Mayer’s hematoxylin, dehydrated, and mounted. Finally, the slides were examined by Olympus light microscopy. Normal kidney glomeruli were used as positive control and negative control was conducted by omitting application of the primary antibody.

The protein expression of TMTC4 appeared as a homogeneous signal in the cytoplasm. Stained slides were examined to identify the cellular localization of immunoreactivity and were evaluated using H-score as we previously described [31]. Briefly, the percentage of positive tumor cells was scored from 1% to 100% and the intensity of expression in positive cells was scored 0–3 (0 equals negative, 1 equals weak, 2 equals moderate, and 3 equals strong signal).

### 4.6. Statistical Analysis

Data were verified and analyzed using SPSS 20.0 (SPSS Inc., Chicago, IL, USA). Data means, standard deviations, medians, ranges, and percentages were calculated. The correlation between TMTC4 expression and the histopathologic characters of PCa was assessed using Kruskal–Wallis H and Mann–Whitney U tests. The expression of the protein was also compared between BPH and PCa groups using the Mann–Whitney test. The receiver operating characteristic (ROC) curve was assembled for TMTC4 protein expression in PCa and BPH and analyzed as the area under the curve (AUC), standard error (SE) and 95% confidence interval (CI). The sensitivity, specificity, and positive and negative predictive values (PPV and NPV) were calculated. Data significance with regards to control group was considered at *p*-value less than 0.05.

## 5. Conclusions

To the best of our knowledge, this is the first study to report the expression pattern of TMTC4 in PCa cells and tissue specimens and highlight its ability to differentiate between the tumor and BPH tissues with high sensitivity and specificity, a finding that can be carried over to clinical practice after further validation. We also found that TMTC4 is overexpressed in PCa at different grades, suggesting an important early role in carcinogenesis. The diagnostic utility and the possible role of TMTC4 in disease progression and relapse of PCa warrant further studies. 

## Figures and Tables

**Figure 1 cells-10-01029-f001:**
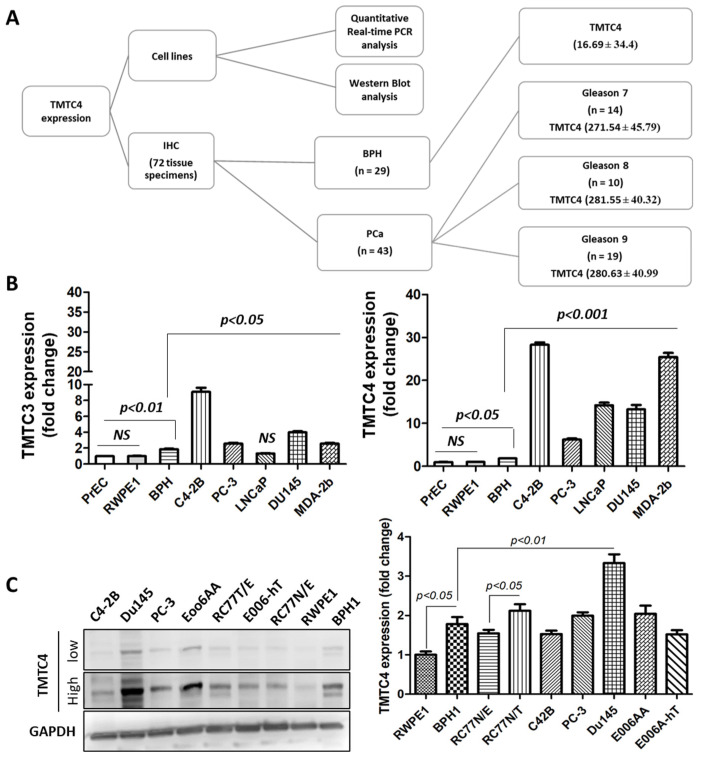
TMTC4 study design and its expression in prostate cancer cells. (**A**) The flowchart of the experimental design of the current study. (**B**) Real-time PCR was performed to determine the expression of *tmtc3* and *tmtc4* in PCa and normal cells using gapdh as a housekeeping gene. Fold change was calculated regarding PEC and RWPE1 cells. (**C**) Sixty µg protein lysate was resolved on SDS-PAGE using TMTC4 antibody. GAPDH antibody was used as a housekeeping protein. Data significance was considered at *p*-value < 0.05 regarding normal and BPH cells, respectively. Experiments repeated at least twice.

**Figure 2 cells-10-01029-f002:**
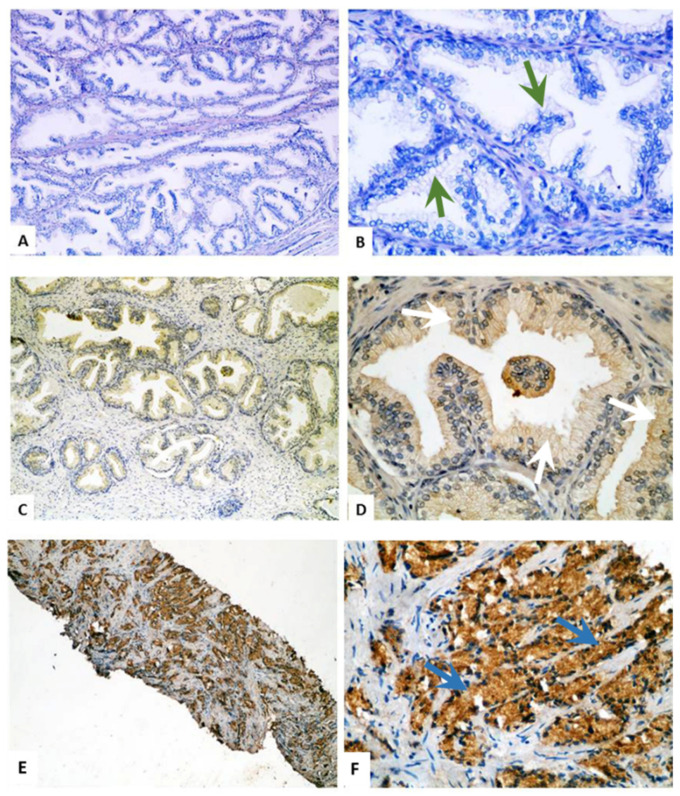
Immunohistochemistry of TMTC4 expression in BPH and prostate cancer patients: (**A**,**B**) TMTC4 staining in BPH showing negative expression in prostatic glands (green arrows); (**C**,**D**) mild cytoplasmic expression of TMTC4 in BPH (white arrows); and (**E**,**F**) strong cytoplasmic expression of TMTC4 in PCa tissues at Gleason score 3 (small glands but still separated from each other by stroma as shown by blue arrows). Magnification: 40× (**A**,**C**,**E**); and 400× (**B**,**D**,**F**).

**Figure 3 cells-10-01029-f003:**
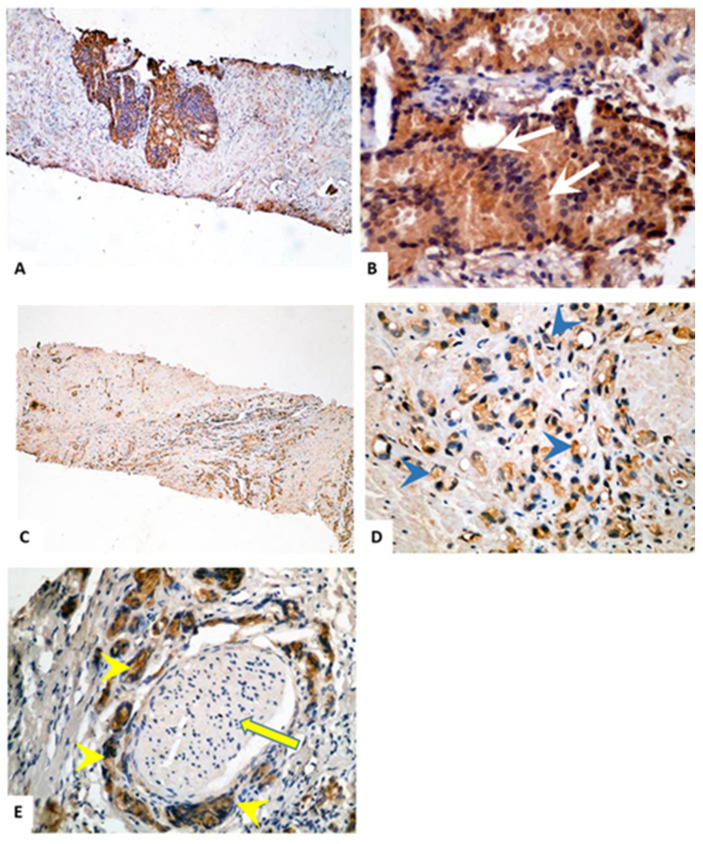
Immunohistochemistry of TMTC4 expression in PCa patients at high Gleason score and perineural invaded tissues: (**A**,**B**) strong cytoplasmic protein expression of TMTC4 in PCa at Gleason score 4 (fused glands with no intervening stroma as depicted by white arrows); (**C**,**D**) strong cytoplasmic staining of TMTC4 in Gleason pattern 5 shows single cells without glandular lumen formation as depicted by arrowhead; and (**E**) strong expression of TMTC4 in perineural invasion of malignant prostatic glands (the yellow arrow indicates the nerve and the arrow heads pointing to malignant glands surrounding the nerve). Magnification: 40× (**A**,**C**); and 400× (**B**,**D**,**E**).

**Figure 4 cells-10-01029-f004:**
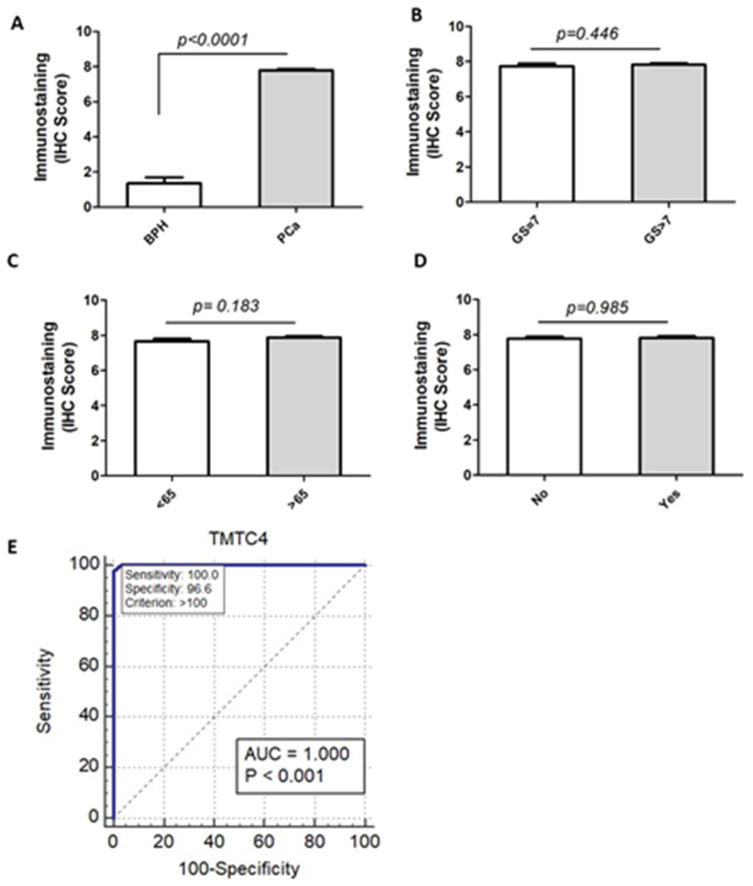
Immunohistoscore of TMTC4 expression in PCa tissues and its sensitivity and specificity: (**A**) immunohistoscore (H-score) of TMTC4 in PCa (*n* = 43) compared to BPH (*n* = 29) tissues; (**B**–**D**) H-sore of the protein based on Gleason score, age, and perineural invasion, respectively; and (**E**) ROC curve analysis for H- score based on TMTC4 protein expression in PCa and BPH tissue specimens. Significance of data was considered at *p* < 0.05.

**Figure 5 cells-10-01029-f005:**
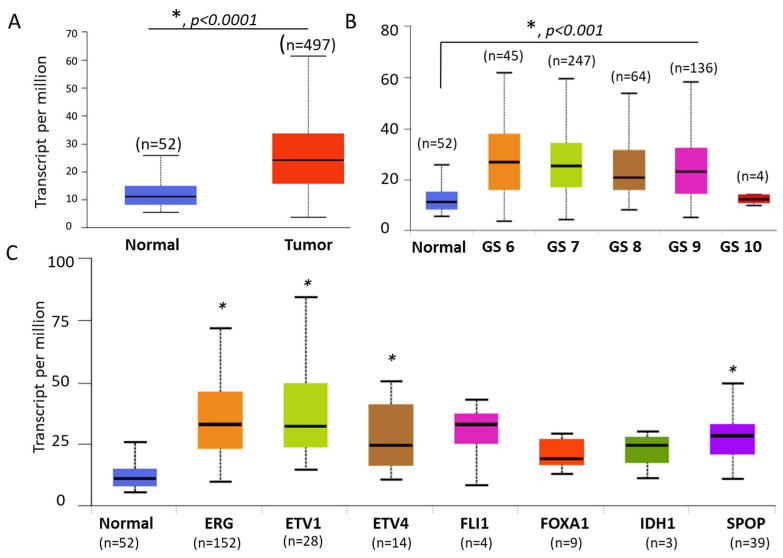
Upregulation of TMTC4 transcript in tissues of patients with PCa using TCGA available data. Available data were retrieved from TCGA, as indicated in the methods. (**A**) Expression of TMTC4 in tumor (*n* = 497) compared to normal (52) tissue specimens. (**B**,**C**) Differential expression of TMTC4 transcripts according to Gleason Score (GS) and molecular features of PCa. * depicts significance at *p* < 0.01.

**Table 1 cells-10-01029-t001:** Description of patients with benign prostatic hyperplasia (BPH) and prostate cancer (PCa).

Variables	Frequency	Percent (%)
**BPH**		
Cases (N)	29	100
Age (average ± SD)	67.3 ± 7.7	
PSA (ng/mL)	6.9 ± 5.6	
**PCa**		
Cases (N)	43	100
Age (average ± SD)	70.3 ± 9.3	
PSA (ng/mL)	14.7 ± 8.6	
**Gleason score**		
7	14	32.5
8	10	23.3
9	19	44.2
**ISUP group**		
2	10	23.3
3	4	9.3
4	10	23.3
5	19	44.2
**Perineural invasion**		
Positive	16	−37.2
negative	27	−62.8

**Table 2 cells-10-01029-t002:** Expression patterns of TMTC4 in BPH and PCa cases.

Variables	TMTC4 Expression (Mean ± SD) (Percent × Intensity)	*p*-Value *
**BPH**	16.69 ± 34.4	<0.001 ^2^
**PCa**	278.12 ± 41.54	
**PCa based on Gleason score**		
7	271.54 ± 45.79	0.88 ^1^
8	281.55 ± 40.32	
9	280.63 ± 40.99	
ISUP Grade		
2	284.00 ± 35.02	0.62 ^1^
3	247.50 ± 60.75	
4	279.70 ± 42.01	
5	280.63 ± 40.99	
**Perineural invasion**		
Positive	275.31 ± 42.87	0.97 ^2^
negative	279.78 ± 41.47	

^1^ Kruskal–Wallis H test. ^2^ Mann–Whitney U test.

**Table 3 cells-10-01029-t003:** ROC Curve analysis for TMTC4 protein expression H-score in PCa compared to BPH tissue specimens.

TMTC4 Marker Cutoff Point	Sensitivity	Specificity	Positive Predictive Value	Negative Predictive Value	AUC	*p*-Value
≥100	100%	96.55%	97.7%	100%	1.0	<0.001

## Data Availability

The generated data of the current study are available on request.

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
