# Peer review of "Transmembrane and Tetratricopeptide Repeat Containing 4 Is a Novel Diagnostic Marker for Prostate Cancer with High Specificity and Sensitivity"

_cells, 2021, doi:10.3390/cells10051029_

Round 1

Reviewer 1 Report

In the manuscript "Transmembrane and tetratricopeptide repeat containing 4 is a novel disagnostic marker for prostate cancer with specificity and sensitivty", the authors attempt to connect expression of TMTC to prostate cancer formation.

The manuscript is of importance as prostate cancer is the cause for the second most male related cancer deaths.

The authors determine TMTC4 expression at the protein and mRNA levels. Using well-established techniques make these observations.

All figures are required.

The methods used in manuscripts are well established and enough detail is given to replicate if need.

While loss of function experiments would have benefited this manuscript, this reviewer understands that the authors intention was to identify a diagnostic marker for prostate cancer and not determine the function of TMTC4 in prostate cancer.

Author Response

Reviewer 1

Comments and Suggestions for Authors

In the manuscript "Transmembrane and tetratricopeptide repeat containing 4 is a novel disagnostic marker for prostate cancer with specificity and sensitivty", the authors attempt to connect expression of TMTC to prostate cancer formation.

The manuscript is of importance as prostate cancer is the cause for the second most male related cancer deaths.

The authors determine TMTC4 expression at the protein and mRNA levels. Using well-established techniques make these observations.

All figures are required.

The methods used in manuscripts are well established and enough detail is given to replicate if need.

While loss of function experiments would have benefited this manuscript, this reviewer understands that the authors intention was to identify a diagnostic marker for prostate cancer and not determine the function of TMTC4 in prostate cancer.

Our response: We would like to thank the reviewer for his/her great comments about our study. We complied with the Reviewer’s comments and we would like to reinstate that the main aim of this study is to identify TMTC4 as a diagnostic biomarker for prostate cancer and further studies are needed to determine its biological significance. 

Reviewer 2 Report

This study conducted by Rania Makbul et al. aimed to explore whether TMTC4 expression has a diagnostic role in PCa through the evaluation of its expression patterns at various grades of the disease and comparing that to the expression of the protein in tissues collected from patients with benign hyperplasia.

Comments and suggestions:

  • A flowchart with the study,s design would be welcomed for readers.
  • In the Introduction section, I suggest adding short data about the definition of biomarkers and Prostate cancer biomarkers in clinical use and in development. The authors also state: “Because of tumor heterogeneity, the histologic diagnosis of PCa is challenging. This is because it depends on finding a constellation of features, alongside the absence of others, rather than relying on a single finding” It is not clear what they have intentionally the authors to mention in this sentence. Given that not all readers are specialists in pathological anatomy, I suggest adding additional explanations or reformulating the phrase.
  • There are many other novel protein biomarkers for PCa. Why did the authors decide to study only this one? In addition, TMTC4 is not at all very new as it is acclaimed by the authorities. There are a plethora of studies on this topic.
  • Legends for figures 2 and 3 should be rewritten and clearly explained, using arrows to show the pathological changes, so that non-speciality readers can understand these photos. (the journal is open access)
  • What is the clinical importance for the potential therapeutic management of prostate cancer of this study?
  • What are the limitations and strong points of this study?

 The paper looks informative and would be useful for the research community. Consider revision accordingly.

Author Response

Reviewer 2

This study conducted by Rania Makbul et al. aimed to explore whether TMTC4 expression has a diagnostic role in PCa through the evaluation of its expression patterns at various grades of the disease and comparing that to the expression of the protein in tissues collected from patients with benign hyperplasia.

Comments and suggestions:

A flowchart with the study,s design would be welcomed for readers.

Our response: We included a new flowchart of the study’s design and presented it as Figure 1A

  • In the Introduction section, I suggest adding short data about the definition of biomarkers and Prostate cancer biomarkers in clinical use and in development. The authors also state: “Because of tumor heterogeneity, the histologic diagnosis of PCa is challenging. This is because it depends on finding a constellation of features, alongside the absence of others, rather than relying on a single finding” It is not clear what they have intentionally the authors to mention in this sentence. Given that not all readers are specialists in pathological anatomy, I suggest adding additional explanations or reformulating the phrase.
  • Our response: We thank the Reviewer’s comments for raising this comment. Accordingly, we added new Reference 6 which reviewing the definition prostate cancer biomarkers in clinical use and development.

In addition, we provided new paragraph which provides more information to the audience as seen below:

“Although several histological diagnostic criteria are described for PCa diagnosis, they still have certain limitations. Moreover, most of these criteria may also be presented in benign mimickers. This explains the challenge of diagnosis of PCa especially in needle biopsy. In addition, some unusual variants of PCa such as foamy carcinoma has bland nuclear features. However, inflamed benign glands, atrophy and atypical basal cell hyperplasia can occasionally display features seen in PCa. One of the most important criteria in diagnosing PCa is the absence of basal cells around the malignant glands [7]. Morphological distinction of basal cells from stromal fibroblasts is often difficult using H&E staining. Furthermore, basal cells may be inconspicuous in some benign glands. Thus, immunohistochemical confirmation of PCa is necessary in certain cases and is traditionally based on the immunoreactivity of basal cell markers (p63 and CK5/6) and cytoplasmic expression of α-methylacyl-CoA racemase (AMACR) [7].” 

  • There are many other novel protein biomarkers for PCa. Why did the authors decide to study only this one? In addition, TMTC4 is not at all very new as it is acclaimed by the authorities. There are a plethora of studies on this topic.

Our response: The field of biomarker discovery is still growing to meet the inter- and intraindividual variations. Although several markers are validated and have been used in clinic, but they have some limitations. We selected TMTC4 to be used as a diagnostic marker with a marked specificity and sensitivity in tissues not in the blood. We acknowledge that TMTC4 is not new as we included the most updated references. However, to the best of our knowledge, there is no available reports on TMTC4 in prostate cancer.

  • Legends for figures 2 and 3 should be rewritten and clearly explained, using arrows to show the pathological changes, so that non-speciality readers can understand these photos. (the journal is open access)

Our response: We complied with the Reviewer suggestions and we updated the legends for figure 2 & 3 and included arrows in figures to make it clear.

Figure 2. Immunohistochemistry of TMTC4 expression in BPH and prostate cancer patients. A &B: TMTC4 staining in BPH showing negative expression in prostatic glands (green arrows). C&D: mild cytoplasmic expression of TMTC4 in BPH (white arrows). E&F: strong cytoplasmic expression of TMTC4 in PCa tissues at Gleason score 6 (small glands but still separated from each other by stroma as shown by blue arrows). Magnification was 40X (A, C &E) and 400X (B, D & F).

Figure 3. Immunohistochemistry of TMTC4 expression in PCa patients at high Gleason score and perineural invaded tissues. A&B: Strong cytoplasmic protein expression of TMTC4 in PCa at Gleason score 8 (fused glands with no intervening stroma as depicted by white arrows). C&D: Strong cytoplasmic staining of TMTC4 in Gleason pattern 10 shows single cells without glandular lumen formation as depicted by arrowhead. E: strong expression of TMTC4 in perineural invasion of malignant prostatic glands (the yellow arrow indicates the nerve and the arrow heads pointing to malignant glands surrounding the nerve). Magnification was 40X (A & C) and 400X (B, D & E).

  • What is the clinical importance for the potential therapeutic management of prostate cancer of this study?

Our response: The potential therapeutic management of PCa in our study is implicated in high diagnostic ability for PCa patients, as histologic PCa mimickers list is wide. In certain situations, PCa may be subtle and overlooked in needle biopsy. Reliable diagnosis of carcinomas will ensure proper management. If TMTC4 can be targeted, it would be very beneficial because it is expressed at all grades so most of patients could benefit from this therapy.

  • What are the limitations and strong points of this study?

Our response: We thank the Reviewer for highlighting these points and we, therefore, added new paragraph at the end of the discuss as shown below:

“Our current study has compelling findings because of 1) the high protein expression of TMTC4 in human tissue specimens was confirmed in PCa as compared to BPH tissue samples with high sensitivity and specificity, 2) the protein expression of TMTC4 was strong in all grades of PCa, and 3) the possibility of using this protein in PCa biopsies as a unique marker. However, the limitations of our study include 1) the number of cases understudy is limited, extended study with more cases is recommended, 2) the expression of this protein in premalignant condition such as prostatic intraepithelial neoplasia (low and high PIN) was not investigated, 3) its expression in other PCa mimickers should be explored to confirm its utility as diagnostic marker for PCa, and 4) the protein expression should be tested in various histologic types of PCa especially those which are difficult to be diagnosed due to its histological similarity of benign conditions.”

 The paper looks informative and would be useful for the research community. Consider revision accordingly.

Our response: We greatly appreciated the positive comments from the Reviewers and we would like also to thank the editor/editorial board for giving us this opportunity to improve the current version of the manuscript.

The Authors,

Round 2

Reviewer 2 Report

The manuscript is revised accordingly. I endorse publication.